# Impact of Special Drying Schemes on Color Stability of Mangoes with Different Maturity Degrees

**DOI:** 10.3390/foods11050656

**Published:** 2022-02-23

**Authors:** Alioune Diop, Jean-Michel Méot, Mathieu Léchaudel, Frédéric Chiroleu, Nafissatou Diop Ndiaye, Christian Mertz, Mady Cissé, Marc Chillet

**Affiliations:** 1CIRAD, UMR QualiSud, 7, Chemin de l’IRAT, La Réunion, 97410 Saint-Pierre, France; marc.chillet@cirad.fr; 2ITA, Route des Pères Maristes, Hann-Dakar BP-2765, Senegal; ndiop@ita.sn (N.D.N.); christian.mertz@cirad.fr (C.M.); 3CIRAD, UMR QualiSud, 34398 Montpellier, France; jean-michel.meot@cirad.fr; 4Qualisud, Univ Montpellier, CIRAD, Institut Agro, Avignon Université, Univ de La Réunion, 34398 Montpellier, France; mathieu.lechaudel@cirad.fr (M.L.); frederic.chiroleu@cirad.fr (F.C.); 5CIRAD, UMR QualiSud, Guadeloupe, 97130 Capesterre-Belle-Eau, France; 6CIRAD, UMR PVBMT, 7 Chemin de l’IRAT, La Réunion, 97410 Saint-Pierre, France; 7Laboratoire d’Electrochimie et des Procédés Membranaires (LEPM), ESP-UCAD, Dakar 10200, Senegal; mady.cisse@ucad.edu.sn

**Keywords:** mango, maturity stage, ripening, dried mangoes, color, quality

## Abstract

A previous study demonstrated that the color of 4 mm mango slices is altered very slightly by drying for 5 h at 60 °C, 30% RH and 1 m/s. The objectives of this complementary study were to determine the impact of various drying procedures encountered in the drying units on color alterations of sulfite-free mango slices from heterogeneous raw material due to variable maturity degrees of mangoes. Drying procedures with various temperature/humidity/duration combinations were performed to analyze their effects on the color of natural dried mangoes according to the degree of fruit maturity. They were dried at an air speed of 1.0 m/s for 5 h according to 3 schemes: standard drying (SD) at 60 °C and 30% RH; wet drying (WD) for 1 h at 60 °C and 60% RH, followed by 4 h SD; and finally, hot drying (HD) for 4 h SD, followed by 1 h at 80 °C and 30% RH. The color of the mango slices was analyzed before and after drying. SD preserves the color of fresh mangoes very well, whatever their maturity stage. A relatively slow drying onset corresponding to WD has a highly adverse impact, which becomes greater as the degree of maturity increases. There is already significant browning on mangoes with near-optimum quality (L* = 75; H* = 92). Applying high temperature at the end of the drying procedure (HD) for 20% of the time has a more limited adverse impact with immature mangoes that are the most sensitive. Linear regressions were assessed to represent the relationships of color differences between drying schemes according to mango maturity degrees. These statistical models showed a significant increase in color degradation in the case of WD and a decrease in color differences in the case of HD with the advance in fruit maturity.

## 1. Introduction

The color of natural, sulfite-free mangoes changes during storage until they become undesirable and unsellable [1]. Currently, they are kept at managed temperatures by many operators, and consumers are advised to store them in the refrigerator. As with the orange sweet potato [2], the color observed is the result of an accumulation of reactions progressing as the fruit matures, from paring–cutting and racking, drying and then storage. Diop et al. [3] showed that a “good drying procedure” caused very little alteration to fresh mango pulp color, regardless of its color or its origin. Production unit operators often point out that this is not the case under real conditions, in particular with the presence of overly dark products. The two generally recognized explanations are, firstly, limited heating power and ventilation, which lead to a slow drying onset during which the surface of the mango pieces remains saturated with water, and, consequently, a development of enzymatic browning reactions on these pieces [4]. Secondly, a high end-of-drying temperature, whether through ignorance or the need to accelerate water diffusion, leads to the development of Maillard reactions [5].

These explanations are derived from a combination of knowledge about water transfers and reactions that take place during drying. If we reproduce the three-phase schematic of the drying behavior of a biological product [6], the intensive conditions in the product and on its surface can be described as follows. The mango pieces are heated during “phase 0”, also known as the “warming-up phase”. The surface temperature of the products is low and gradually reaches the wet temperature, which depends on the temperature and relative humidity of the drying air [7]. Thereafter, external transfers are limiting (“phase 1”, also known as the “constant rate drying phase”). The surface a_w_ is close to 1 [7], and the temperature is equal to the wet bulb temperature. Internal transfers are limiting when the surface a_w_ drops below 1. This is “phase 2”, also known as the “falling rate phase”. In this phase, the surface temperature increases and the water content decreases until it reaches values which depend on the characteristics of the air and on the product composition [6].

During phase 1 of drying, the high water content is favorable for enzymatic browning reactions [8], while it is excessive for Maillard reactions [9]. The temperature has little effect on the specific reaction rate of the enzyme, and above 50 °C the enzymes are progressively inactivated [10]. Browning is often observed in drying, though under extreme properly selected conditions, the main effect can be enzyme inactivation [11]. During phase 2, the water activity of the material rapidly drops below 1, especially on the surface. This area is particularly sensitive since: (i) it is its color which is perceived; (ii) it contains cells which are fully damaged; and (iii) it is in contact with oxygen in the air. Enzymatic browning rates fall rapidly, while conditions become favorable for Maillard reactions and oxidation of the carotenoids. The very many publications relating to enzymatic activities and Maillard reactions lead us to believe that modeling approaches would be straightforward, especially since the water and heat transfers have been modeled [12,13]. Yet this is certainly not the case. For enzymatic browning reactions, it is primarily the deactivation kinetics which have been studied, and for Maillard reactions, it is primarily the kinetics of change in concentration of intermediate compounds [14]. The information provided is interesting, but we are lacking some essential information to simulate color alterations.

Furthermore, the variability of the raw material is not often considered when analyzing the effect of drying conditions on color changes. However, in mango-drying units, the raw material comes from various origins in terms of varieties, maturity degrees and other pre- and postharvest parameters influencing mango composition [15]. This latter factor, as documented by Diop et al. [3], causes differences in the content of substrates involved in color degradation reactions. In this case, the increase with the maturity degree of the contents of reducing sugars [16] and amino acids [17], which are the substrates of the Maillard reaction, and that of the carotenoid contents, leads to more intensive oxidation during convective drying [18]. The same applies to the substrates and enzymes involved in enzymatic browning (respectively, polyphenols, PPO and POD) [19,20].

The aim of this study was to evaluate the consequences on color of the two most common defects of the drying process for non-sulfited mangoes, in relation to their maturity degrees. Three different drying schemes (by managing time, temperature and relative humidity of air) were applied to mango slices from fruits of different degrees of maturity. Analyses of covariance (ANCOVA) were performed to test whether the relations between the color indices measured on the dried slices and those measured on the same slices before drying were influenced by the drying type. The differences in color index between drying schemes were discussed in relation to changes in maturity degree.

## 2. Materials and Methods

### 2.1. Plant Materials

First, 60 cv. Cogshall mangoes were harvested at the yellow-point stage (i.e., 54 mangoes for the study and 6 supplementary fruits if problems were encountered during ripening) and sampled according to the experimental design (Appendix A). The yellow-point specific stage of the Cogshall variety corresponds to the start of the climacteric crisis, which is recognized through yellowing of the mango’s apex [21]. At harvest, each mango was tagged individually. They were immersed in a bath of chlorinated water (0.6% sodium hypochlorite) for 5 min, rinsed 3 times in clean water and then ripened at 20 °C. For this trial, fruits were ripened to generate variable maturity degrees via successive withdrawals spaced 2 days apart. Three withdrawals of mangoes were performed at 2, 4 and 6 days after harvest. At each withdrawal, 18 mangoes were chosen, to form 3 batches of 6 fruits to undergo 3 different types of drying. Upon sampling, mangoes were stored at 12 °C, in order to limit changes in quality and maturity after sampling [22]. Mangoes kept for 6 days at 20 °C reached maturity as desired for processing into natural dried mangoes. The 7th day after harvest, the 3 formed batches each contained 6 mangoes ripened 2, 4 and 6 days at 20 °C, i.e., a total of 18 mangoes per batch. A specific drying scheme was applied per batch as described in the following paragraph.

### 2.2. Mango Drying

For each batch, each of the 18 mangoes were peeled and then cut using a slicer. Four slices per mango with a homogeneous thickness of 4 mm were selected to be dried. The remainder of each mango pulp was crushed separately to perform physicochemical analyses. Each slice was identified to indicate its mango of origin, along with its ripening–storage pathway.

All the three drying schemes were carried out by using a forced-convection electric dryer, with a steam-injection- and air-extraction-based relative humidity regulation system. This dryer was entirely designed and manufactured by CIRAD’s mechanical and process engineering teams (Appendix A). For all the tests, the mango slices were placed in a single layer on the drying racks and the air flow circulating in parallel to slices was at a fixed speed of 1.0 m/s. Batch 1 underwent standard drying (SD), batch 2 wet drying (WD) and batch 3 hot drying (HD). This experiment was carried out one time on a total of 72 slices of mangoes for each drying scheme. SD was performed at constant temperature and relative humidity, 60 °C and 30% RH, respectively. The duration applied was 5 h. WD and HD were similar to SD, with modifications in the first and last hour, respectively. For WD, the first hour was at 60% RH instead of 30% RH (SD). For HD, the last hour was at 80 °C instead of 60 °C (SD). The modifications made were weak compared to the drying process defects observed in some drying units.

### 2.3. Quality Measurement

A color measurement (average of 5 flashes) was taken in the center of each slice, using a Minolta CR-410 (Tokyo, Japan). The measurements were taken before and after drying, to be able to quantify color variations. The L*, a* and b* data were converted into L*, H* and C*. The C*-based color alterations are not presented, due to the fact that this index is not a good predictor of changes in carotenoid contents for the Cogshall variety [23], nor for browning or color alterations during drying [3].

Chemical analyses were performed on the residual pulp of each crushed mango separately. The soluble solids content (°Brix) was measured using a portable refractometer (PAL-α, Atago, Tokyo, Japan). Titratable acidity (TA) and pH were measured using an automatic titrator (TitroLine, Schott Instruments, Mainz, Germany) with 1 g of pulp, 10 mL of water and titrated with a 0.05 N solution of NaOH.

### 2.4. Statistical Analyses and Modeling

The statistical analyses were performed using the R software [24]. The means of the physicochemical parameters for each ripening time were compared using a Tukey test to evaluate the effect of this factor, i.e., the ripening time, on the changes in the quality characteristics of the mangoes. Then, linear regressions were performed between the color indices (L* dried, H* dried) of the dried mangoes, and those measured on the same fresh slices (L* fresh, H* fresh), as a function of drying type. The effect of drying type was analyzed by an analysis of covariance (ANCOVA), with the indices L* fresh and H* fresh as independent variables, L* dried and H* dried as dependent variables and the factor “drying type” (SD, HD or WD) as covariable. For each relation index* dried = f (index*fresh; drying type), a comparison of the slopes was performed to evaluate the differences between the regression lines. A comparison of the intercepts was performed if the slopes were not significantly different (*p* < 0.05). Homogenous groups were formed as a function of these differences (*p* < 0.05).

Six linear models were obtained, corresponding to one linear relation per color index CI* (L* and H*) and per drying type (SD, WD and HD):CI*_dried_,_SD_,_model_ = A_dried_,_SD_ + B_dried_,_SD_ CI*_fresh_,_SD_,_exp_(1)
CI*_dried_,_WD_,_model_ = A_dried_,_WD_ + B_dried_,_WD_ CI*_fresh_,_WD_,_exp_(2)
CI*_dried_,_HD_,_model_ = A_dried_,_HD_ + B_dried_,_HD_ CI*_fresh_,_HD_,_exp_(3)
where CI*_dried, drying type, exp_ is the color index (L* or H*) of the dried mangoes predicted by the linear regression models; A_CI*_ and B_CI*_ are, respectively, the intercepts and slopes of the linear models as a function of the drying type; and CI*_fresh, drying type, exp_ are the experimental values of the index L* or H* measured on the fresh slices before each drying type.

The differences $\Delta {\mathrm{CI}}_{i}^{*}$ (ΔL* and ΔH*) between the regression line of standard drying and the regression lines for hot drying and wet drying were calculated for each index using the above linear models obtained for the three drying schemes and data for wet drying and for hot drying as detailed in Equations (4) and (5).
∆CI*_SD-HD_ = (A_dried_,_SD_ − A_dried_,_HD_) + (B_dried_,_SD_ − B_dried_,_HD_) CI*_fresh_,_HD_,*_exp_*(4)
∆CI*_SD-WD_ = (A_dried_,_SD_ − A_dried_,_WD_) + (B_dried_,_SD_ − B_dried_,_WD_) CI*_fresh_,_WD_,*_exp_*(5)

Then, linear regressions of $\Delta {\mathrm{CI}}_{i}^{*}$ as a function of the physicochemical parameters *PC_i_* (PC_1_ = TSS, PC_2_ = pH, PC_3_ = TA, PC_4_ = TSS/TA) were performed. Differences between these regression lines were also analyzed by covariance analyses. Pearson’s correlation coefficients between $\Delta {\mathrm{CI}}_{i}^{*}$ and the physicochemical parameters were calculated according to the drying schemes using the cor.test() function. Interpretation of those results made it possible to link color alterations to maturity degrees of mangoes.

## 3. Results and Discussion

### 3.1. Effect of Ripening on Fresh Mango Quality

The ripening time to achieve a good degree of maturity for drying was 6 days from mango harvesting. This short time is understandable, since mangoes harvested at the yellow-point stage have already begun their climacteric crisis on the tree. Since the preclimacteric phase is non-existent [25], the evolution of the physicochemical characteristics during ripening is very quick. Figure 1 shows the evolution of the physicochemical and colorimetric parameters during ripening. The successive withdrawals on days 2, 4 and 6 provided mangoes of increasing maturity degrees, with a significant increase (*p* < 0.05) in °Brix and pH and a significant fall in titratable acidity. According to Yashoda et al. [16], the increase in °Brix is due to the conversion of starch into simple sugars. Figure 1a shows the stabilization in °Brix from the 4th day. In parallel, the fall in citric and malic acid contents causes the fall in titratable acidity (Figure 1b) and the increase in pH (Figure 1c) [26]. On the 6th day, the titratable acidity has practically finished dropping, since it has reached a value close to 0. The color of the mangoes’ flesh has also changed during ripening, with a significant decrease in L* and H*, without reaching stabilization. However, this would not have been long in coming, as the values of the color indexes reached were close to the optima at 20 °C described by Diop et al. [3]. The evolution of the physicochemical parameters and their timing were similar to those reported by Noiwan et al. [27] on the Nam Dok Mai variety. In their experience, the °Brix reached a maximum at 8 days, while acidity, firmness and color continued to change until the 12th day. This suggests different dynamics over the course of ripening between the evolution of the primary metabolites (carbohydrates: starch and sugars) and secondary metabolites (carotenoids, aroma compounds, etc.), specific to the variety. Pandit et al. [28] stated that the expressions of genes associated with primary and secondary metabolisms were not synchronized. Levels of expression appear to depend on the growth-ripening stage and on biotic stress factors (ethylene) and abiotic stress factors (growth conditions).

The creation of the different maturity degrees therefore generated three similar batches, i.e., batch 1, subjected to SD; batch 2, to WD; and batch 3, to HD, and each of these three batches comprised fruits of three different withdrawals.

### 3.2. Effect of Drying Type and Maturity on Color Changes of Natural Dried Mangoes

Figure 2 represents the linear regressions between the color indexes L* and H* measured on the dried slices under the three drying conditions as a function of the same indices measured on the same fresh slices. Drying type has a significant effect on the relations (*p* < 0.05) (Figure 2a,b). The analyses of covariance and comparisons of the slopes and/or intercepts made it possible to specify these differences.

SD did not alter the luminance of the fresh mangoes (Figure 2a). The regression line overlaps with the first bisector, indicating that the luminance value after drying is equal to that of fresh mango. The results of Diop et al. [3] were reproduced with other mangoes: drying at 60 °C/30% RH/1.0 m/s for 5 h only causes very slight alterations to the luminance of fresh mangoes. The envelope curves of the 95% confidence intervals of the regression lines of the hot and wet drying procedures are completely underneath the first bisector, indicating that the luminance after drying is lower than before drying. Similar results were reported by Izli et al. [29] and Zou et al. [30].

The fall in luminance caused by wet drying can be attributed primarily to enzymatic browning [4]. The higher relative humidity at the drying onset increases the duration of phase 1 of drying, characterized by a limiting external exchange. During this phase, the water activity of the surface remains close to 1, and the product temperature is close to the wet air temperature. This value is 39.7 °C for air at 60 °C and 30% RH, and 50.4 °C for air at 60 °C and 60% RH. Maintaining sufficient a_w_ for the enzymatic browning reactions to progress smoothly and a low reduction in the specific reaction rate of the enzyme [10] emerge as predominant factors upon inactivation of the enzyme at 50.4 °C.

The slope of the linear regression line L_dried_ = f(L_fresh_) of the wet drying procedure is significantly different (*p <* 0.05) from those of standard and hot drying (Figure 2a). The slope of more than 1 and the remoteness of the regression line from the first bisector at low luminance values show the higher sensitivity of the ripest mangoes (low L*) to enzymatic browning. Figure 3, showing the evolution of ΔL* (L*standard drying—L*studied drying) as a function of the maturity indices, confirms that increasing maturity is associated with greater browning during wet drying.

The correlation coefficients of the regressions lines of ΔL* as a function of pH and titratable acidity (Table 1), are greater than 0.8 (*p* < 10^−15^). This high correlation could be due to the increased activity of PPO with increasing pH [31], and consequently the falling acidity. The browning sensitivity of the ripest mangoes may also come from their higher richness in polyphenols [32] and the smaller presence of vitamin C [33], the concentration of which decreases with ripening [20]. The lower correlation with °Brix can be attributed to the fact that this parameter quickly reached its maximum during ripening and then stabilized (Figure 1). Conversely, color, °Brix/AT and pH continued to change significantly, with a broader variation range (Figure 1c–f). The same order of magnitude for variations in °Brix as a function of maturity and variability within a drying batch [34] represents another reason for the limited correlation between ΔL* and °Brix.

Hot drying causes a decrease in luminance L* over the whole region studied, of around 3 units (Figure 2a). According to the covariance analyses, there is a significant difference in slopes between this drying type and the wet one (*p* < 0.05). Differentiated from standard drying only by a higher temperature during the last hour, the browning is attributable to Maillard reactions [35], with a high degree of progression at medium to low a_w_, associated with high temperatures. At the end of hot drying, the water activity is low and the temperature high, which corresponds to conditions favorable for the Maillard reaction. Applying a temperature of 80 °C over a greater fraction of drying or using a temperature above 80 °C would have caused greater browning [36].

The regression lines (L*dried as a function of L*fresh) for standard and hot drying have the same slope but significantly different intercepts (*p* < 0.05). This is unlike the results of Corzo and Alvarez [37]; it seems that the least ripe mangoes are the most sensitive to browning (Figure 3) during hot drying. This is surprising, since the least ripe mangoes, in comparison to ripe mangoes: (i) are poorer in reducing sugars and proteins [38]; (ii) their lower pH is less favorable for Maillard reactions [4]; (iii) are richer in vitamin C; and (iv) have more intact intracellular structures. However, the low a_w_ values reached, which occurs more quickly for mangoes with low sugar content [39], and the high temperature of hot drying, could cause the slightly more intense Maillard reaction for these immature mangoes.

All the drying types cause alteration of the index H* toward lower hues, as is shown by the position of the experimental points in Figure 2b. This is consistent with the observations of Caparino et al. [40] and Nyangena et al. [41] during drying of mangoes using various techniques. This fall is apparently due to discoloration due to breakdown of the carotenoids [30]. Wet drying does not present any significant differences from standard drying, whether in terms of its 95% envelope curve or comparison of the slopes and intercepts of the regression lines for H* dried = f (H* fresh) (Figure 2). The alterations to H* caused by the wet and standard drying procedures are only very slightly dependent on mango maturity degree (Figure 3; Table 1). Figure 3, representing ΔH* (H* standard drying—H* studied drying) as a function of maturity indices, shows a practically constant trajectory of ΔH* for wet drying. Hot drying provides very different results from the other two drying procedures, although the averages of the ΔH* are similar: alteration of H* is highly dependent on maturity degree (Table 1); it is low for ripe mangoes and high for immature mangoes. Since there is a large increase in carotenoid content during ripening, the increase in reaction rate as a function of temperature [42] should be more visible in ripe mangoes, i.e., those rich in carotenoids, than in less mature mangoes.

## 4. Conclusions

The results obtained agree with the empirical knowledge of fruit drying operators: optimal quality cannot be obtained with a too slow drying onset due to insufficient heating power, an overloaded mango dryer or the application of a too high temperature at the end of drying performed to reduce drying duration.

The wet drying scheme (WD) caused the most intense browning, with the greatest decrease in L* especially for ripe mangoes. The hot drying (HD) scheme induced color degradation of all samples, but this effect was smaller than the WD one and more significant for immature mangoes. The standard drying (SD) was the best drying scheme, maintaining the color of the mango slices perfectly, with an identical L* before and after drying, in particular.

Relationships of color differences between drying schemes according to mango maturity degrees were accurately represented by linear regressions. These statistical models showed a significant positive correlation of color degradation with the advance in maturity in the case of WD and a significant negative one in the case of HD. This highlights once again the major impact of these drying conditions.

The joint dependence of color stability on fruit maturity but also on drying conditions has been clearly demonstrated. The study of color, which is the primary criterion for purchasing mangoes, was relevant because its modification according to the applied conditions made it possible to synthesize in a concrete way the result of browning reactions, which are various, complex and very difficult to study.

## Figures and Tables

**Figure 1 foods-11-00656-f001:**
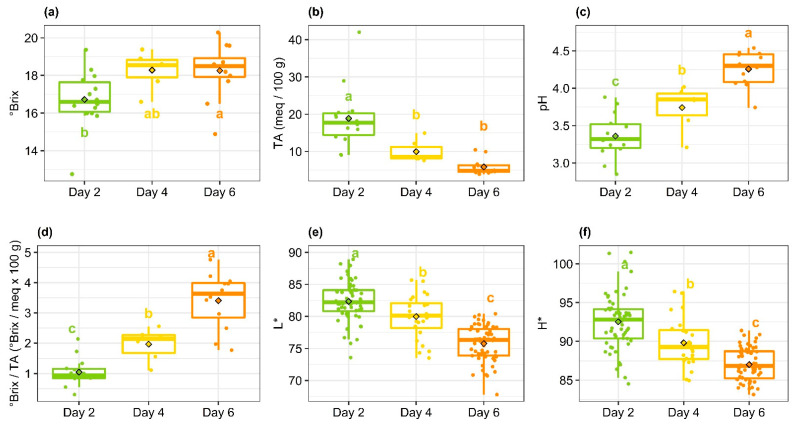
Box plots of °Brix (**a**), titratable acidity: TA (**b**), pH (**c**), °Brix/titratable acidity (**d**), L* (**e**) and H* (**f**) as a function of ripening time at 20 °C. Dots represent individual measurements. Different letters indicate significant differences in mean values among three days with *p* < 5%.

**Figure 2 foods-11-00656-f002:**
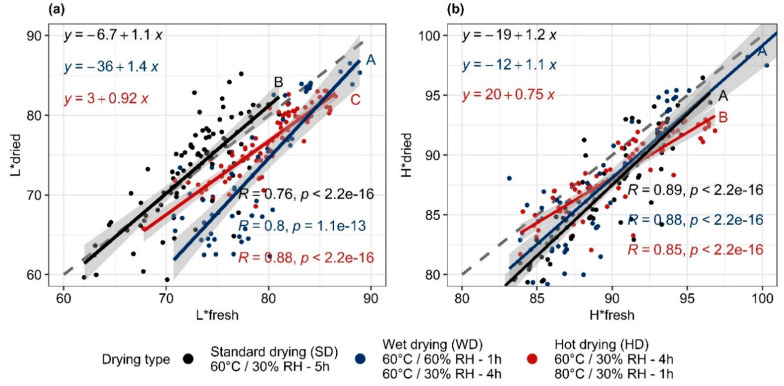
Linear regressions of color indices L* (**a**) and H* (**b**) measured on dried slices as a function of the same color indices measured on the same slices before drying, for each drying scheme. Grey dashed line: first bisector; dots: experimental data; solid lines with grey envelopes: regression lines with 95% confidence intervals. Different letters indicate significantly different regression lines at 5% threshold.

**Figure 3 foods-11-00656-f003:**
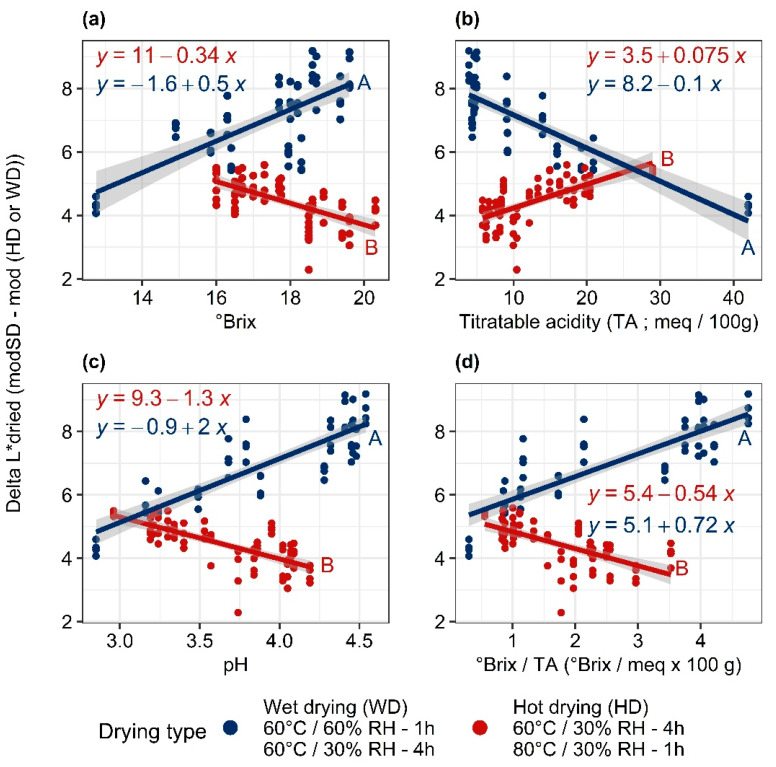
Regressions of ΔL* as a function of physicochemical indices (°Brix (**a**), titratable acidity: TA (**b**), pH (**c**), °Brix/titratable acidity (**d**)) under wet and hot drying procedures. Solid lines with grey envelopes: regression lines with 95% confidence intervals. Different letters indicate significantly different regression lines at 5% threshold.

**Table 1 foods-11-00656-t001:** Pearson’s correlation coefficients between the ΔColor index of standard drying and wet or hot drying schemes for indices L* and H* and the physicochemical (*PC_i_*) indicators. Statistical significance between slopes of the regression lines presented in Figure 2 is indicated.

ΔColor Indices	ΔL*			ΔH*		
Drying Type/PC_i_ Characteristics	Wet Drying (WD)	Hot Drying (HD)		Wet Drying (WD)	Hot Drying (HD)	
°Brix	0.71	−0.63	***	−0.66	−0.67	***
TA (meq/100 g)	−0.83	0.66	***	0.90	0.71	***
pH	0.86	−0.71	***	−0.88	−0.77	***
°Brix/TA	0.84	−0.64	***	−0.80	−0.67	***

*** Slopes are significantly different at 0.1% threshold.

## Data Availability

The data presented in this study are available on request from the corresponding author.

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
