# Peer review of "Impact of Special Drying Schemes on Color Stability of Mangoes with Different Maturity Degrees"

_foods, 2022, doi:10.3390/foods11050656_

Round 1
Reviewer 1 Report
Revised article "Impact of Special Drying Schemes on Color Stability of Man-2 goes with Different Maturity Degrees"
This article seems to me to be relevant for the dehydrated mango production industry, since color is a very important quality parameter for consumers.
The methodology described was very well planned and explained.
The description of the equipment used for the determination of all the parameters of interest is missing. It is necessary to at least make a descriptive table.
A graphic summary will help a better understanding of the methodology and the results obtained.
The graphic description of the dryer used is necessary and of interest for the development of the manuscript.
In figure 1 make the lines thinner to better appreciate the dispersion of the data.
Table 1 . Why were the SD results not integrated into the table?
line 180 - 188. there are editing/programming errors in the cited references
The introduction describes very well the process and the stages of a drying kinetics however, there was not a single drying kinetics as a result! It is necessary to put the drying and drying rate graphs for each of the methods used SD, WD and HD
The conclusions seem very poor to me. There is a lot of relevant information within the article that can be summarized and inferred in the conclusions.
For everything else, it seems to me that the article was well done
Author Response
Please see the responses in the attached file.

Reviewer 2 Report
General remarks
- The authors did not explain the exact purpose of the research. It is not known whether the aim of the research was to find the most favorable ripening phase and to optimize the drying conditions? In the introduction, it is theoretically shown that the chemical composition changes during maturation. The chemical composition can determine the color, and that is clear.
- But the authors used two independent variables: the degree of maturity and 3 different drying modes. But they did not justify the purpose of modifying the drying parameters: The authors theoretically show that, in the first state of drying, too high humidity of the drying air increases the development of enzymatic browning reactions on the surface (line 47-48). Nevertheless, they increase the humidity in the first hour of drying, which must slow down drying and additionally raise the temperature of the material. Under such conditions, it is not possible for the first drying period to occur!! And the browning process needs to be more intense. Wasn't it better to reduce the humidity of the drying air? In turn, the second modification of the drying conditions - increasing the temperature at the end of drying, leads to the development of Maillard reactions (Line 50, 51). Did the authors want this effect? After all, oxidation reactions accelerate when partial removal of water facilitates direct access to bioactive molecules. An increase in temperature accelerates the kinetics of biochemical and chemical reactions, thus also oxidation
- They did not take into account the parameters that can determine the color, regardless of the degree of maturity and drying parameters, and may be a consequence of these two variables: The authors did not specify the final water content, which can have a significant impact on the color. The less water, the higher the concentration of chemicals - the color must be more intense, and vice versa. Materials significantly different in water content should not be compared, and if they are, another independent variable - water content must be taken into account. During drying, the material may shrink. The shrinkage itself can intensify the color, because a given amount of dyes is contained in a smaller volume.
The discussion and presentation of the results is correct - no comments
Detailed comments
Introduction
Line 52 - 64 - These messages are not related to the scope of the research performed. And they should be removed. During the drying process, the temperature of the material and the loss of mass were not controlled, so it was not possible to determine whether the first drying period occurred at all (in my opinion, it does not occur, because the resistance to mass movement in undamaged mango tissue is very high).
There is no justification as to why the assessment of the color of the product is important.
Figure 1 d. There is an invalid unit on the value axis. Should be (oBrix / meq * 100 g)
A statement in conclusion: “The phenomena and reactions potentially involved are numerous and complex, mainly dependent on numerous parameters, in particular initial composition, the structure of the material and the intensive conditions generated by the process (temperature, water content / water activity, partial oxygen pressure, etc.), and especially the distinguished terms, are not justified in the research conducted by the authors.
Author Response

(The authors gave the same response as above.)

Round 2
Reviewer 2 Report
The explanations and corrections introduced by the authors are enough to make my doubts disappear.